# An algebraic approach to intertwined quantum phase transitions in the Zr isotopes

N. Gavrielov[1,2][*]

**1** Center for Theoretical Physics, Sloane Physics Laboratory, Yale University, New Haven,
Connecticut 06520-8120, USA
**2** Racah Institute of Physics, The Hebrew University, Jerusalem 91904, Israel
* noam.gavrielov@yale.edu

December 14, 2022

**Group**
ICGTMP

## Abstract

**The algebraic framework of the interacting boson model with configuration mixing is em-
ployed to demonstrate the occurrence of intertwined quantum phase transitions (IQPTs)
in the $_{40}$Zr isotopes with neutron number 52–70. The detailed quantum and classical
analyses reveal a QPT of crossing normal and intruder configurations superimposed on
a QPT of the intruder configuration from U(5) to SU(3) and a crossover from SU(3) to
SO(6) dynamical symmetries.**

## 1 Introduction

Quantum phase transitions [1–3] are qualitative changes in the structure of a physical system that occur as a function of one (or more) parameters that appear in the quantum Hamiltonian describing the system. In nuclear physics [4], we vary the number of nucleons and examine mainly two types of quantum phase transitions (QPTs). The first describes shape phase transitions in a single configuration, denoted as Type I. When interpolating between two shapes, for example, the Hamiltonian can be written as a sum of two parts

$$\hat{H} = (1 - \xi)\hat{H}_1 + \xi\hat{H}_2 \, , \tag{1}$$

with $\xi$ the control parameter. As we vary $\xi$ with nucleon number from 0 to 1, the equilibrium shape and symmetry of the Hamiltonian vary from those of $\hat{H}_1$ to those of $\hat{H}_2$. QPTs of this type have been studied extensively in the framework of the interacting boson model (IBM) [4–7]. One example of such QPT is the $_{62}$Sm region with neutron number 84–94, where the shape evolves from spherical to axially-deformed, with a critical point at neutron number 90.

The second type of QPT occurs when the ground state configuration changes its character, typically from normal to intruder type of states, denoted as Type II QPT. In such cases, the Hamiltonian can be written in matrix form [8]. For two configurations $A$ and $B$ we have

$$\hat{H} = \left[ \begin{array}{cc} \hat{H}_A(\xi_A) & \hat{W}(\omega) \\ \hat{W}(\omega) & \hat{H}_B(\xi_B) \end{array} \right] \, , \tag{2}$$

with $\xi_i$ ($i = A, B$), the control parameter of configuration ($i$), and $\hat{W}$, the coupling between them with parameter $\omega$. QPTs of this type are manifested empirically near (sub-) shell closure, e.g. in the light Pb-Hg isotopes, with strong mixing between the configurations [9, 10].

Recently, we have introduced a new type of phase-transitions in even-even [11, 12] and odd-mass [13] nuclei called intertwined quantum phase transitions (IQPTs). The latter refers to a scenario where as we vary the control parameters ($\xi_A, \xi_B, \omega$) in Eq. (2), each of the Hamiltonians $\hat{H}_A$ and $\hat{H}_B$ undergoes a separate and clearly distinguished shape-phase transition (Type I), and the combined Hamiltonian simultaneously experiences a crossing of configurations $A$ and $B$ (Type II).

## 2 Theoretical framework

A convenient framework to study the different types of QPTs together is the extension of the IBM to include configuration mixing (IBM-CM) [14–16].

### 2.1 The interacting boson model with configuration mixing

The IBM for a single shell model configuration has been widely used to describe low-lying quadrupole collective states in nuclei in terms of $N$ monopole ($s^\dagger$) and quadrupole ($d^\dagger$) bosons, representing valence nucleon pairs. The model has U(6) as a spectrum generating algebra, where the Hamiltonian is expanded in terms of its generators, $\{s^\dagger s, s^\dagger d_\mu, d_\mu^\dagger s, d_\mu^\dagger d_{\mu'}\}$, and consists of Hermitian, rotational-scalar interactions which conserve the total number of $s$- and $d$-

64   bosons $\hat{N} = \hat{n}_s + \hat{n}_d = s^\dagger s + \sum_\mu d_\mu^\dagger d_\mu$ . The boson number is fixed by the microscopic interpre-
65   tation of the IBM [17] to be $N = N_\pi + N_\nu$, where $N_\pi$ ($N_\nu$) is the number of proton (neutron)
66   particle or hole pairs counted from the nearest closed shell.

67       The solvable limits of the model correspond to dynamical symmetries (DSs) associated
68   with chains of nested sub-algebras of U(6), terminating in the invariant SO(3) algebra. In the
69   IBM there are three DS limits

$$
\text{U(6)} \supset \begin{cases} \text{U(5)} \supset \text{SO(5)} \supset \text{SO(3)}, \\ \text{SU(3)} \supset \text{SO(3)}, \\ \text{SO(6)} \supset \text{SO(5)} \supset \text{SO(3)}. \end{cases} \tag{3}
$$

70   In a DS, the Hamiltonian is written in terms of Casimir operators of the algebras of a given
71   chain. In such a case, the spectrum is completely solvable and resembles known paradigms of
72   collective motion: spherical vibrator [U(5)], axially symmetric [SU(3)] and $\gamma$-soft deformed
73   rotor [SO(6)]. In each case, the energies and eigenstates are labeled by quantum numbers
74   that are the labels of irreducible representations (irreps) of the algebras in the chain. The
75   corresponding basis states for each of the chains (3) are

$$\text{U(5)}: \quad |N, n_d, \tau, n_\Delta, L\rangle, \tag{4a}$$

$$\text{SU(3)}: \quad |N, (\lambda, \mu), K, L\rangle, \tag{4b}$$

$$\text{SO(6)}: \quad |N, \sigma, \tau, n_\Delta, L\rangle, \tag{4c}$$

76   where $N, n_d, (\lambda, \mu), \sigma, \tau, L$ label the irreps of U(6), U(5), SU(3), SO(6), SO(5) and SO(3),
77   respectively, and $n_\Delta, K$ are multiplicity labels.

78       An extension of the IBM to include intruder excitations is based on associating the different
79   shell-model spaces of 0p-0h, 2p-2h, 4p-4h, ... particle-hole excitations, with the corresponding
80   boson spaces with $N, N+2, N+4, ...$ bosons, which are subsequently mixed [15,16]. For two
81   configurations the resulting IBM-CM Hamiltonian can be transcribed in a form equivalent to
82   that of Eq. (2)

$$\hat{H} = \hat{H}_A^{(N)} + \hat{H}_B^{(N+2)} + \hat{W}^{(N,N+2)} . \tag{5}$$

83   Here, the notations $\hat{\mathcal{O}}^{(N)} = \hat{P}_N^\dagger \hat{\mathcal{O}} \hat{P}_N$ and $\hat{\mathcal{O}}^{(N,N')} = \hat{P}_N^\dagger \hat{\mathcal{O}} \hat{P}_{N'}$, stand for an operator $\hat{\mathcal{O}}$, with
84   $\hat{P}_N$, a projection operator onto the $N$ boson space. The Hamiltonian $\hat{H}_A^{(N)}$ represents the $N$
85   boson space (normal $A$ configuration) and $\hat{H}_B^{(N+2)}$ represents the $N+2$ boson space (intruder
86   $B$ configuration).

## 2.2   Wave functions structure

88   The eigenstates $|\Psi; L\rangle$ of the Hamiltonian (5) with angular momentum $L$, are linear combina-
89   tions of the wave functions, $\Psi_A$ and $\Psi_B$, in the two spaces $[N]$ and $[N+2]$,

$$|\Psi; L\rangle = a |\Psi_A; [N], L\rangle + b |\Psi_B; [N+2], L\rangle , \tag{6}$$

90   with $a^2 + b^2 = 1$. We note that each of the components in Eq. (6), $|\Psi_A; [N], L\rangle$ and $|\Psi_B; [N+2], L\rangle$,
91   can be expanded in terms of the different DS limits with its corresponding boson number in
92   the following manner

$$|\Psi_i; [N_i], L\rangle = \sum_\alpha C_\alpha^{(N_i, L)} |N_i, \alpha, L\rangle, \tag{7}$$

93   where $N_A = N$ and $N_B = N + 2$, and $\alpha = \{n_d, \tau, n_\Delta\}, \{(\lambda, \mu), K\}, \{\sigma, \tau, n_\Delta\}$ are the quantum
94   numbers of the DS eigenstates. The coefficients $C_\alpha^{(N,L)}$ give the weight of each component

in the wave function. Using them, we can calculate the wave function probability of having definite quantum numbers of a given symmetry in the DS bases, Eq. (7), for its $A$ or $B$ parts

$$\text{U(5)}: \quad P_{n_d}^{(N_i,L)} = \sum_{\tau,n_\Delta} [C_{n_d,\tau,n_\Delta}^{(N_i,L)}]^2, \qquad \text{SO(6)}: \quad P_\sigma^{(N_i,L)} = \sum_{\tau,n_\Delta} [C_{\sigma,\tau,n_\Delta}^{(N_i,L)}]^2, \qquad (8a)$$

$$\text{SU(3)}: \quad P_{(\lambda,\mu)}^{(N_i,L)} = \sum_K [C_{(\lambda,\mu),K}^{(N_i,L)}]^2, \qquad \text{SO(5)}: \quad P_\tau^{(N_i,L)} = \sum_{n_d,n_\Delta} [C_{n_d,\tau,n_\Delta}^{(N_i,L)}]^2. \qquad (8b)$$

Here the subscripts $i = A,B$ denote the different configurations, i.e., $N_A = N$ and $N_B = N + 2$. Furthermore, for each eigenstate (6), we can also examine its coefficients $a$ and $b$, which portray the probability of the normal-intruder mixing. They are evaluated from the sum of the squared coefficients of an IBM basis. For the U(5) basis, we have

$$P_a^{(N_A,L)} \equiv a^2 = \sum_{n_d,\tau,n_\Delta} |C_{n_d,\tau,n_\Delta}^{(N_A,L)}|^2; \qquad P_b^{(N_B,L)} \equiv b^2 = \sum_{n_d,\tau,n_\Delta} |C_{n_d,\tau,n_\Delta}^{(N_B,L)}|^2. \qquad (9)$$

where the sum goes over all possible values of $(n_d, \tau, n_\Delta)$ in the $(N_i, L)$ space, $i = A,B$, and $a^2 + b^2 = 1$.

## 2.3 Geometry

To obtain a geometric interpretation of the IBM is we take the expectation value of the Hamiltonian between coherent (intrinsic) states [5, 18] to form an energy surface

$$E_N(\beta,\gamma) = \langle \beta,\gamma;N|\hat{H}|\beta,\gamma;N\rangle \ . \qquad (10)$$

The $(\beta,\gamma)$ of Eq. (10) are quadrupole shape parameters whose values, $(\beta_{eq}, \gamma_{eq})$, at the global minimum of $E_N(\beta,\gamma)$ define the equilibrium shape for a given Hamiltonian. The values are $(\beta_{eq} = 0)$, $(\beta_{eq} = \sqrt{2}, \gamma_{eq} = 0)$ and $(\beta_{eq} = 1, \gamma_{eq}$ arbitrary) for the U(5), SU(3) and SO(6) DS limits, respectively. Furthermore, for these values the ground-band intrinsic state, $|\beta_{eq}, \gamma_{eq};N\rangle$, becomes a lowest weight state in the irrep of the leading subalgebra of the DS chain, with quantum numbers $(n_d = 0)$, $(\lambda,\mu) = (2N,0)$ and $(\sigma = N)$ for the U(5), SU(3) and SO(6) DS limits, respectively.

For the IBM-CM Hamiltonian, the energy surface takes a matrix form [19]

$$E(\beta,\gamma) = \begin{bmatrix} E_A(\beta,\gamma;\xi_A) & \Omega(\beta,\gamma;\omega) \\ \Omega(\beta,\gamma;\omega) & E_B(\beta,\gamma;\xi_B) \end{bmatrix}, \qquad (11)$$

where the entries are the matrix elements of the corresponding terms in the Hamiltonian (2), between the intrinsic states of each of the configurations, with the appropriate boson number. Diagonalization of this two-by-two matrix produces the so-called eigen-potentials, $E_\pm(\beta,\gamma)$.

## 2.4 QPTs and order parameters

The energy surface depends also on the Hamiltonian parameters and serves as the Landau potential whose topology determines the type of phase transition. In QPTs involving a single configuration (Type I), the ground state shape defines the phase of the system, which also identifies the corresponding DS as the phase of the system. Such Type I QPTs can be studied using a Hamiltonian as in Eq. (1), that interpolates between different DS limits (phases) by varying its control parameters $\xi$. The order parameter is taken to be the expectation value of the $d$-boson number operator, $\hat{n}_d$, in the ground state, $\langle \hat{n}_d \rangle_{0_1^+}$, and measures the amount of deformation in the ground state.

In QPTs involving multiple configurations (Type II), the dominant configuration in the ground state defines the phase of the system. Such Type II QPTs can be studied using a Hamiltonian as in Eq. (5), that interpolates between the different configurations by varying its control parameters $\xi_A, \xi_B, \omega$. The order parameters are taken to be the expectation value of $\hat{n}_d$ in the ground state wave function, $|\Psi; L = 0_1^+\rangle$, and in its $\Psi_A$ and $\Psi_B$ components, Eq. (6), denoted by $\langle \hat{n}_d \rangle_{0_1^+}$, $\langle \hat{n}_d \rangle_A$ and $\langle \hat{n}_d \rangle_B$, respectively. The shape-evolution in each of the configurations $A$ and $B$ is encapsulated in $\langle \hat{n}_d \rangle_A$ and $\langle \hat{n}_d \rangle_B$, respectively. Their sum weighted by the probabilities of the $\Psi_A$ and $\Psi_B$ components $\langle \hat{n}_d \rangle_{0_1^+} = a^2 \langle \hat{n}_d \rangle_A + b^2 \langle \hat{n}_d \rangle_B$, portrays the evolution of the normal-intruder mixing.

# 3 QPTs in the Zr isotopes

Along the years, the $Z \approx 40$, $A \approx 100$ region was suggested by many works to have a ground state that is dominated by a normal spherical configuration for neutron numbers 50–58 and by an intruder deformed configuration for 60 onward. This dramatic change in structure is explained in the shell model by the isoscalar proton-neutron interaction between non-identical nucleons that occupy the spin-orbit partner orbitals $\pi 1 g_{9/2}$ and $\nu 1 g_{7/2}$ [20]. The crossing between configurations arises from the promotion of protons across the Z=40 subsell gap. The interaction energy results in a gain that compensates the loss in single-particle and pairing energy and a mutual polarization effect is enabled. Therefore, the single-particle orbitals at higher intruder configurations are lowered near the ground state normal configuration, which effectively reverses their order.

## 3.1 Model space

Using the framework of the IBM-CM, we consider $^{90}_{40}$Zr as a core and valence neutrons in the 50–82 major shell. The normal $A$ configuration corresponds to having no active protons above $Z = 40$ sub-shell gap, and the intruder $B$ configuration corresponds to two-proton excitation from below to above this gap, creating 2p-2h states. Therefore, the IBM-CM model space employed in this study, consists of $[N] \oplus [N + 2]$ boson spaces with total boson number $N = 1, 2, \ldots 8$ for $^{92-106}$Zr and $\bar{N} = \bar{7}, \bar{6}$ for $^{108,110}$Zr, respectively, where the bar over a number indicates that these are hole bosons.

## 3.2 Hamiltonian and $E2$ transitions operator

In order to describe the spectrum of the Zr isotopes, we take a Hamiltonian that has a form as in Eq. (5) with entries

$$\hat{H}_A(\epsilon_d^{(A)}, \kappa^{(A)}, \chi) = \epsilon_d^{(A)} \hat{n}_d + \kappa^{(A)} \hat{Q}_\chi \cdot \hat{Q}_\chi \,, \tag{12a}$$

$$\hat{H}_B(\epsilon_d^{(B)}, \kappa^{(B)}, \chi) = \epsilon_d^{(B)} \hat{n}_d + \kappa^{(B)} \hat{Q}_\chi \cdot \hat{Q}_\chi + \kappa'^{(B)} \hat{L} \cdot \hat{L} + \Delta_p \,, \tag{12b}$$

where the quadrupole operator is given by $\hat{Q}_\chi = d^\dagger s + s^\dagger \tilde{d} + \chi (d^\dagger \times \tilde{d})^{(2)}$, and $\hat{L} = \sqrt{10}(d^\dagger \tilde{d})^{(1)}$ is the angular momentum operator. Here $\tilde{d}_m = (-1)^m d_{-m}$ and standard notation of angular momentum coupling is used. The off-set energy between configurations $A$ and $B$ is $\Delta_p$, where the index $p$ denotes the fact that this is a proton excitation. The mixing term in Eq. (5) between configurations $(A)$ and $(B)$ has the form [14–16] $\hat{W} = \omega [ (d^\dagger \times d^\dagger)^{(0)} + (s^\dagger)^2 ] + \text{H.c.}$, where H.c. stands for Hermitian conjugate. The parameters are obtained from a fit, elaborated in the appendix of Ref. [12].

The $E2$ operator for two configurations is written as $\hat{T}(E2) = e^{(A)} \hat{Q}_\chi^{(N)} + e^{(B)} \hat{Q}_\chi^{(N+2)}$, with $\hat{Q}_\chi^{(N)} = \hat{P}_N^\dagger \hat{Q}_\chi \hat{P}_N$ and $\hat{Q}_\chi^{(N+2)} = \hat{P}_{N+2}^\dagger \hat{Q}_\chi \hat{P}_{N+2}$. The boson effective charges $e^{(A)}$ and $e^{(B)}$ are

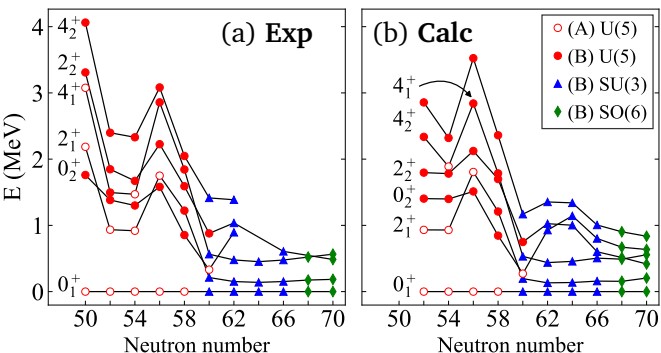

Figure 1: Comparison between (a) experimental and (b) calculated energy levels $0_1^+, 2_1^+, 4_1^+, 0_2^+, 2_2^+, 4_2^+$. Empty (filled) symbols indicate a state dominated by the normal $A$ configuration (intruder $B$ configuration), with assignments based on Eq. (9). The symbol[ ●, ▲, ♦ ], indicates the closest dynamical symmetry [U(5), SU(3), SO(6)] to the level considered, based on Eq. (8). Note that the calculated values start at neutron number 52, while the experimental values include the closed shell at 50. References for the data can be found in [12].

determined from the $2^+ \rightarrow 0^+$ transition within each configuration [12], and $\chi$ is the same parameter as in the Hamiltonian (12).

For the energy surface matrix (11), we calculate the expectation values of the Hamiltonians $\hat{H}_A$ (12a) and $\hat{H}_B$ (12b) in the intrinsic state of Section 2.3 with $N$ and $N+2$ bosons respectively, and a non-diagonal matrix element of the mixing term $\hat{W}$ between them. The explicit expressions can be found in [12].

# 4 Results

In order to understand the change in structure of the Zr isotopes, it is insightful to examine the evolution of different properties along the chain.

## 4.1 Evolution of energy levels

In Fig. 1, we show a comparison between selected experimental and calculated levels, along with assignments to configurations based on Eq. (9) and to the closest DS based on Eq. (8), for each state. In the region between neutron number 50 and 56, there appear to be two configurations, one spherical (seniority-like), ($A$), and one weakly deformed, ($B$), as evidenced by the ratio $R_{4/2}$, which is $R_{4/2}^{(A)} \cong 1.6$ and $R_{4/2}^{(B)} \cong 2.3$ at at 52–56. From neutron number 58, there is a pronounced drop in energy for the configuration ($B$) states and at 60, the two configurations exchange their role, indicating a Type II QPT. At this stage, the $B$ configuration appears to undergo a U(5)-SU(3) Type I QPT, similarly to case of the Sm region [14, 21, 22]. Beyond neutron number 60, the $B$ configuration is strongly deformed, as evidenced by the small value of the excitation energy of the state $2_1^+$, $E_{2_1^+} = 139.3$ keV and by the ratio $R_{4/2}^{(B)} = 3.24$ in $^{104}$Zr. At still larger neutron number 66, the ground state band becomes $\gamma$-unstable (or triaxial) as evidenced by the close energy of the states $2_2^+$ and $4_1^+$, $E_{2_2^+} = 607.0$ keV, $E_{4_1^+} = 476.5$ keV, in $^{106}$Zr, and especially by the results $E_{4_1^+} = 565$ keV and $E_{2_2^+} = 485$ keV for $^{110}$Zr of Ref. [23], a signature of the SO(6) symmetry. In this region, the $B$ configuration undergoes a crossover from SU(3) to SO(6).

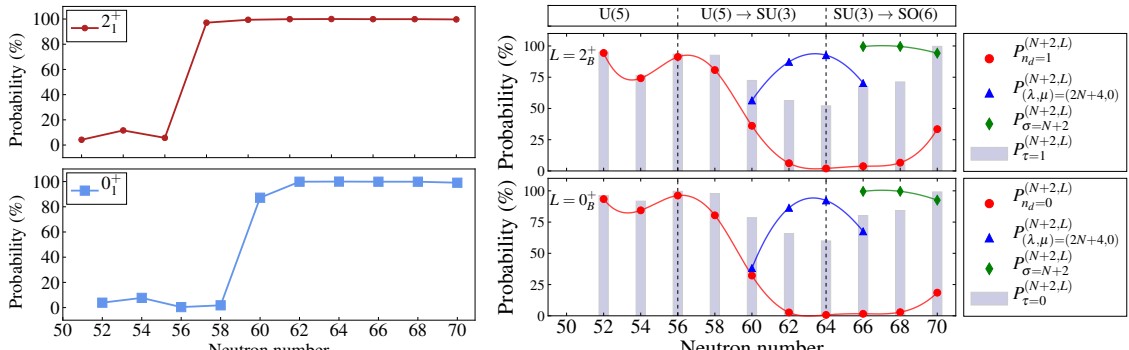

Figure 2: Left panels: percentage of the wave functions within the intruder B-configuration [the $b^2$ probability in Eq. (6)], for the ground $0_1^+$ (bottom) and excited $2_1^+$ (top) states in $^{92-110}$Zr. Right panels: evolution of symmetries for the lowest $0^+$ (bottom) and $2^+$ (top) state of configuration $B$ along the Zr chain. Shown are the probabilities of selected components of U(5) (●), SU(3) (▲), SO(6) (◆) and SO(5) (▬), obtained from Eq. (8). For neutron numbers 52–58 (60–70), $0_B^+$ corresponds to the experimental $0_2^+$ ($0_1^+$) state. For neutron numbers 52–56 (58–70), $2_B^+$ corresponds to the experimental $2_2^+$ ($2_1^+$) state.

## 4.2 Evolution of configuration content

We examine the configuration change for each isotope, by calculating the evolution of the probability $b^2$, Eq. (9), of the $0_1^+$ and $2_1^+$ states. The left panels of Fig. 2 shows the percentage of the wave function within the $B$ configuration as a function of neutron number across the Zr chain. The rapid change in structure of the $0_1^+$ state (bottom left panel) from the normal $A$ configuration in $^{92-98}$Zr (small $b^2$ probability) to the intruder $B$ configuration in $^{100-110}$Zr (large $b^2$ probability) is clearly evident, signaling a Type II QPT. The configuration change appears however sooner in the $2_1^+$ state (top left panel), which changes to configuration $B$ already in $^{98}$Zr, in line with [24]. Outside a narrow region near neutron number 60, where the crossing occurs, the two configurations are weakly mixed and the states retain a high level of purity, especially for neutron number larger than 60.

## 4.3 Evolution of symmetry content

We examine the changes in symmetry of the lowest $0^+$ and $2^+$ states within the $B$ configuration, which undergoes a Type I QPT. In the right bottom panel of Fig. 2 the red dots represent the percentage of the U(5) $n_d = 0$ component in the wave function, $P_{n_d=0}^{(N+2,L=0)}$ of Eq. (8). It is large ($\approx 90\%$) for neutron number 52–58 and drops drastically ($\approx 30\%$) at 60. The drop means that other $n_d \neq 0$ components are present in the wave function and therefore this state becomes deformed. Above neutron number 60, the $n_d = 0$ component drops almost to zero (and rises again a little at 70), indicating the state is strongly deformed. To understand the type of DS associated with the deformation above neutron number 60, we add in blue triangles the percentage of the SU(3) $(\lambda, \mu) = (2N + 4, 0)$ component, $P_{(\lambda,\mu)=(2N+4,0)}^{(N+2,L=0)}$ of Eq. (8) for 60–66. For neutron number 60, it is moderately small ($\approx 35\%$), at neutron number 62 it jumps ($\approx 85\%$) and becomes maximal at 64 ($\approx 92\%$). This serves as a clear evidence for a U(5)-SU(3) Type I QPT. At neutron number 66 the SU(3) $(\lambda, \mu) = (2N+4, 0)$ component it is lowered, and one sees by the green diamonds the percentage of the SO(6) $\sigma = N + 2$ component, $P_{\sigma=N+2}^{(N+2,L=0)}$ of Eq. (8). The latter becomes dominant for 66–70 ($\approx 99\%$), suggesting a crossover from SU(3) to SO(6).

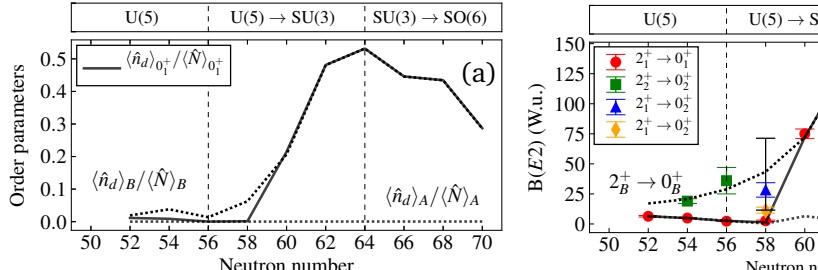

Figure 3: (a) Evolution of order parameters along the Zr chain, normalized (see text). (b) $B(E2)$ values in W.u. for $2^+ \to 0^+$ transitions in the Zr chain. The solid line (symbols ●, ■, ▲, ◆) denote calculated results (experimental results). Dotted lines denote calculated $E2$ transitions within a configuration. The data for $^{94}$Zr, $^{96}$Zr, $^{100}$Zr, $^{102}$Zr and ($^{104}$Zr, $^{106}$Zr) are taken from [25], [26], [27], [28], [29], respectively. For $^{98}$Zr (neutron number 58), the experimental values are from [30] (◆), from [31] (▲), and the upper and lower limits (black bars) are from [24, 27].

In order to further elaborate the Type I QPT within configuration $B$ from U(5) to SU(3) and the subsequent crossover to SO(6), we examine also the evolution of SO(5) symmetry. The gray histograms in the right panel of Fig. 2 depict the probability of the $\tau = 0$ component of SO(5), $P_{\tau=0}^{(N+2,L=0)}$ of Eq. (8), for $0_B^+$. For neutron numbers 52–56, the $0_B^+$ state is composed mainly of a single $(n_d = 0, \tau = 0)$ component, appropriate for a with state good U(5) DS. For neutron number 58, the larger $\tau = 0$ but smaller $n_d = 0$ probabilities imply the presence of additional components with $(n_d \neq 0, \tau = 0)$. For neutron numbers 60–64, the $\tau = 0$ probability decreases, implying admixtures of components with $(n_d \neq 0, \tau \neq 0)$, appropriate for a state with good SU(3) DS. For neutron numbers 66–70, the $\tau = 0$ probability increases towards its maximum value at 70, appropriate for a crossover to SO(6) structure with good SO(5) symmetry.

In the top right panel of Fig. 2 we observe a similar trend for the $2_B^+$ state. For neutron numbers 52–58, it is dominated by a single $(n_d = 1, \tau = 1)$ component. For neutron number 60, $P_{n_d=1}^{(N+2,L=2_B^+)}$ is smaller than $P_{\tau=1}^{(N+2,L=2_B^+)}$, indicating the onset of deformation. For 62–64, $P_{n_d=1}^{(N+2,L=2_B^+)}$ is much smaller than $P_{\tau=1}^{(N+2,L=2_B^+)}$, implying admixtures of components with $(n_d \neq 1, \tau \neq 1)$. For neutron numbers 66–70, $P_{n_d=1}^{(N+2,L=2_B^+)}$ remains small but $P_{\tau=1}^{(N+2,L=2_B^+)}$ increases towards its maximum value at 70.

## 4.4 Evolution of order parameters

The configuration and symmetry analysis of Sections 4.2 and 4.3 suggest a situation of simultaneous occurrence of Type I and Type II QPTs. The order parameters can give further insight to these QPTs. Fig. 3(a) shows the evolution along the Zr chain of the order parameters ($\langle \hat{n}_d \rangle_A$, $\langle \hat{n}_d \rangle_B$ in dotted and $\langle \hat{n}_d \rangle_{0_1^+}$ in solid lines), normalized by the respective boson numbers, $\langle \hat{N} \rangle_A = N$, $\langle \hat{N} \rangle_B = N+2$, $\langle \hat{N} \rangle_{0_1^+} = a^2 N + b^2 (N+2)$. The order parameter $\langle \hat{n}_d \rangle_{0_1^+}$ is close to $\langle \hat{n}_d \rangle_A$ for neutron number 52–58 and coincides with $\langle \hat{n}_d \rangle_B$ at 60 and above. The clear jump and change in configuration content from 58 to 60 indicates a Type II phase transition [8], with weak mixing between the configurations. Configuration $A$ is spherical for all neutron numbers, and configuration $B$ is weakly-deformed for neutron number 52–58. From neutron number 58 to 60 we see a sudden increase in $\langle \hat{n}_d \rangle_B$ that continues towards 64, indicating a U(5)-SU(3) Type I phase transition. Then, we observe a decrease from neutron number 66 onward, due in part to the crossover from SU(3) to SO(6) and in part to the shift from bo-

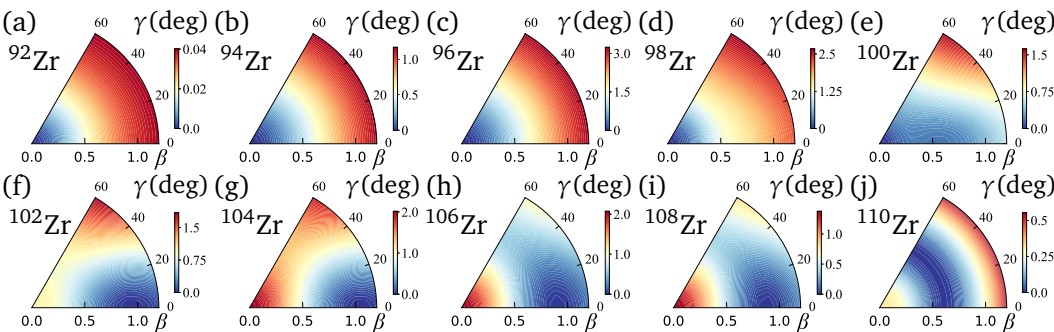

Figure 4: Contour plots in the $(\beta, \gamma)$ plane of the lowest eigen-potential surface, $E_-(\beta, \gamma)$, for the $^{92-110}$Zr isotopes.

son particles to boson holes after the middle of the major shell 50–82. These conclusions are stressed by an analysis of other observables [12], in particular, the $B(E2)$ values. As shown in Fig. 3(b), the calculated $B(E2)$'s agree with the experimental values and follow the same trends as the respective order parameters.

## 4.5 Classical analysis

In Fig. 4, we show the calculated lowest eigen-potential $E_-(\beta, \gamma)$, which is the lowest eigenvalue of the matrix Eq. (11). These classical potentials confirm the quantum results, as they show a transition from spherical ($^{92-98}$Zr), Figs. 4(a)-(d), to a double-minima potential that is almost flat-bottomed at $^{100}$Zr, Fig. 4(e), to prolate axially deformed ($^{102-104}$Zr), Figs. 4(f)-(g), and finally to $\gamma$-unstable ($^{106-110}$Zr), Figs. 4(h)-(j).

# 5 Conclusions and outlook

The algebraic framework of the IBM-CM allows us to examine QPTs using both quantum and classical analyses. We have employed this analysis to the Zr isotopes with $A=92$–110, which exhibit a complex structure that involves a shape-phase transition within the intruder configuration (Type I QPT) and a configuration-change between normal and intruder (Type II QPT), namely IQPTs. This was done by analyzing the energies, configuration and symmetry content of the wave functions, order parameters and $E2$ transition rates, and the energy surfaces. Further analysis of other observables supporting this scenario is presented in [12]. Recently, we have also exemplified the notion IQPTs in the odd-mass $_{41}$Nb isotopes [13] and it would be interesting to examine the notion of IQPTs in other even-even and odd-mass chains of isotopes in the $Z \approx 40$, $A \approx 100$ region and other physical systems.

# Acknowledgments

This work was done in collaboration with F. Iachello (Yale university) and A. Leviatan (Hebrew university).

**Funding information** This work was supported in part by the US-Israel Binational Science Foundation Grant No. 2016032 and the Israel Academy of Sciences for a Postdoctoral Fellowship Program in Nuclear Physics.

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
