# Peer review of "An algebraic approach to intertwined quantum phase transitions in the Zr isotopes"

_SciPost Physics Proceedings_

## Round 1 · Referee Report · Anonymous (Referee 1) · 2023-3-6

Strengths
- The author introduces a novel concept in the application of quantum phase transitions in atomic nuclei.
- The paper is very well written, the text is clear and concise.
- The study uses abundantly Lie algebraic notions and therefore has its place in the proceedings of the Group Theory meeting.
Weaknesses
- This is a parametrised approach to a phenomenon that, in principle, can be described more microscopically (although with tremendous numerical effort) in the context of the shell model.
Report
In this contribution the author reports on a recently introduced novel concept related to quantum phase transitions (QPTs) in atomic nuclei. Based on earlier work by Gilmore and co-workers concerned with general aspects in quantum systems, QPTs were introduced in 1980 in the context of the interacting boson model (IBM) [5,18]. For over two decades this approach remained a standard way for describing shape evolution in nuclei until it was realised [8] that a second type of evolution occurs if, when changing the number of neutrons or protons, the nucleus switches from one to another coexisting configuration with a different shape. The two mechanisms of QPT are referred to as type I and II, respectively.
The novel aspect of the present contribution is to argue that both QPTs can occur simultaneously, giving rise to what has been named an intertwined QPT, or IQPT. In addition, the author shows compelling evidence that this IQPT actually occurs in the zirconium isotopes.
The contribution is an excellently written summary of what has been done so far related to IQPTs. It adds to the work published in [12] by its clarity and conciseness and shows modified, in some cases updated, figures. As the study requires the permanent use of algebraic concepts in the context of the IBM, it has its place in the proceedings of the Group Theory meeting that took place in 2022 in Strasbourg.
I can therefore advise publication of the manuscript. My only comment is about the right panel of Figure 2, where some of the greek characters (tau, sigma, lambda and mu) have disappeared, at least in the PDF file available to me.
The novel aspect of the present contribution is to argue that both QPTs can occur simultaneously, giving rise to what has been named an intertwined QPT, or IQPT. In addition, the author shows compelling evidence that this IQPT actually occurs in the zirconium isotopes.
The contribution is an excellently written summary of what has been done so far related to IQPTs. It adds to the work published in [12] by its clarity and conciseness and shows modified, in some cases updated, figures. As the study requires the permanent use of algebraic concepts in the context of the IBM, it has its place in the proceedings of the Group Theory meeting that took place in 2022 in Strasbourg.
I can therefore advise publication of the manuscript. My only comment is about the right panel of Figure 2, where some of the greek characters (tau, sigma, lambda and mu) have disappeared, at least in the PDF file available to me.
Requested changes
- See my comment on Figure 2 in my report.

Author: Noam Gavrielov on 2023-03-09 [id 3457]
(in reply to Report 1 on 2023-03-06)Dear Referee,
I would like to thank you for your thuoghtful report on the manuscript.
I have corrected figure 2 as you pointed out.
Sincerely,
Dr. Noam Gavrielov

---

## Editorial Decision

resubmitted